# Antiviral Activity of Ethyl Gallate Against Zika Virus: In Vitro and In Silico Studies

**DOI:** 10.3390/ijms262412062

**Published:** 2025-12-15

**Authors:** Yeon-Ji Lee, Nalae Kang, Jun-Ho Heo, Eun-A Kim, Soo-Jin Heo

**Affiliations:** 1Jeju Bio Research Center, Korea Institute of Ocean Science and Technology (KIOST), Jeju 63349, Republic of Korea; leeyj0409@kiost.ac.kr (Y.-J.L.); nalae1207@kiost.ac.kr (N.K.); unknown0713@kiost.ac.kr (J.-H.H.); euna0718@kiost.ac.kr (E.-A.K.); 2Department of Marine Technology and Convergence Engineering, University of Science and Technology (UST), Daejeon 34113, Republic of Korea

**Keywords:** Zika infection, ethyl gallate, antiviral, molecular docking, molecular dynamic simulation

## Abstract

Zika virus (ZIKV) remains a significant global public health concern, and growing resistance to existing antiviral drugs underscores the necessity of developing alternative therapeutic options. In this study, we investigated the inhibitory effects of ethyl gallate against ZIKV using antiviral activity evaluation, molecular docking, and molecular dynamic simulations. Treatment of ZIKV-infected Vero E6 cells with ethyl gallate resulted in dose-dependent suppression of viral infection without inducing cytotoxicity. In addition, ethyl gallate inhibited the increase in the expression of interferon-stimulated genes in ZIKV-infected cells. It also exhibited binding energies of −5.9868, −247.271, and −200.43 kcal/mol for ZIKV envelope, NS3, and RdRp proteins, respectively. Furthermore, the molecular dynamic simulation results showed that the ethyl gallate-NS3 and ethyl gallate-RdRp complexes were more stable than the ethyl gallate-envelope protein complex, suggesting that ethyl gallate has the potential to inhibit ZIKV replication. These findings position ethyl gallate as an antiviral agent with potential against Zika infection.

## 1. Introduction

Zika virus (ZIKV) is an arthropod-borne virus (arbovirus) belonging to the genus *Flavivirus* and family Flaviviridae [1]. The major route of transmission for most flaviviruses is through arthropod vectors, which are typically asymptomatic and cause a relatively benign, mild, febrile illness characterized by fever, arthralgia, headache, malaise, maculopapular rash, and conjunctivitis [2,3]. However, ZIKV infections have also been associated with severe neurological complications such as Guillain–Barré syndrome, meningoencephalitis, myeletis, and congenital microcephaly [4]. Although there is no specific antiviral therapy for Zika virus, since 2017, its prevalence has gradually declined. Nevertheless, it continues to pose a persistent health burden in many global regions, especially Asia and Africa, underscoring the urgent need for the sustained development of effective therapeutics [5].

ZIKV is a single-stranded, enveloped RNA virus that consists of three structural proteins, capsular (C), premembrane (prM), and envelope (E), and seven non-structural proteins: NS1, NS2A, NS2B, NS3, NS4A, NS4B, and NS5. These proteins are important components of its structure, protecting the viral genetic material, identifying host receptors, and supporting viral replication [6].

Nucleoside analogs, the nucleotide analog prodrug sofosbuvir, and the alkaloid emetine have been reported to exhibit anti-ZIKV activity by targeting the RdRp domain within NS5 in certain cells [7,8]. Many studies have observed that some antibiotic and antimalarial drugs possess in vitro and in vivo anti-ZIKV activities [8]. Although various drugs have been studied as potential anti-ZIKV agents, no vaccine or drug has yet been approved to treat or prevent ZIKV infections.

Ethyl gallate, the ethyl ester of gallic acid, is a phenolic compound found in natural plants and is known for its potent antioxidant activity. It has attracted considerable attention because of its diverse pharmacological properties, including anti-inflammatory, antibacterial, and antitumor properties and melanin production inhibition [9,10,11,12]. This study examined the potential antiviral activity of ethyl gallate against ZIKV-infected Vero E6 cells and characterized its structural interactions using molecular docking and molecular dynamic (MD) simulations.

## 2. Results

### 2.1. Anti-ZIKV Effect of Ethyl Gallate in Vero E6 Cells

To determine the antiviral efficacy of ethyl gallate, we assessed viral plaque formation and ZIKV NS1 mRNA expression (Figure 1). Ethyl gallate exhibited no cytotoxicity in Vero E6 cells across the 25–200 μM range (Figure 1a,b) and suppressed ZIKV plaque formation at all tested concentrations (Figure 1c). On the other hand, chloroquine, a well-known antiviral drug, showed clear cytotoxicity at low concentrations—1.95, 3.91, 7.81, and 15.63 μM—reducing cell viability to 83.21%, 74.26%, 65.96%, and 57.04%, respectively (See Appendix A for details). Treatment with ethyl gallate resulted in a gradual reduction in both the number and size of ZIKV plaques, which was evident even at low concentrations (25–50 µM). Viral infection (0.01 multiplicity of infection [MOI]) showed a plaque-forming virus titer of 2670 ± 127 PFU/mL; however, ethyl gallate treatment decreased this to 760 ± 85 and 320 ± 28 PFU/mL at concentrations of 25 and 50 µM, respectively. In addition, complete inhibition of plaque formation was evident in cells exposed to 100 and 200 µM ethyl gallate (Figure 1c). Moreover, viral infection (0.01 MOI) increased the NS1 mRNA expression level to 2,676,567 times that in the Mock group (non-treatment group), whereas ethyl gallate treatment (25, 50, 100, and 200 µM) decreased these levels to 763,658 ± 33,879, 397,989 ± 24,204, 100,016 ± 2975, and 36,192 ± 1321, respectively (Figure 1e). These results demonstrated the potent antiviral activity of ethyl gallate against ZIKV.

### 2.2. Effects of Ethyl Gallate on Interferon (IFN)-Stimulated Gene (ISG) Expression in ZIKV-Infected Vero E6 Cells

The IFN system is essential in immune responses and serves as a frontline defense against viruses [13]. Therefore, we investigated the effects of ethyl gallate on ISG mRNA expression. Using reverse transcription PCR (RT-PCR) analysis, we analyzed the expression levels of genes that block viral entry (IFITM family), inhibit protein translation (IFIT family), induce RNA degradation (OAS), and suppress protein synthesis (PKR) as well as TNFα—a cytokine that regulates inflammatory and immune responses. IFITM1, IFITM3, IFIT1, IFIT2, IFIT3, IFIT5, OAS1, OAS2, PKR (*p* < 0.0001), and TNFα (*p* < 0.0002), which were significantly upregulated in the positive control (PC) group, were downregulated after treatment with ethyl gallate (Figure 2). Therefore, ethyl gallate downregulates ISG expression in ZIKV-infected Vero E6 cells by inhibiting ZIKV entry and translation, degrading viral RNA, and suppressing protein synthesis.

### 2.3. Molecular Docking of Ethyl Gallate with Three Main ZIKV Proteins

To investigate the structural characteristics supporting the antiviral activity of ethyl gallate, we performed a molecular docking analysis of ethyl gallate bound to the major proteins of ZIKV, viz. the envelope protein, NS3 helicase, and RdRp. Ethyl gallate formed complexes with each protein via non-bonded interactions, as indicated by the CDOCKER interaction energies and calculated binding energies (Figure 3, Figure 4 and Figure 5 and Table 1). Specifically, ethyl gallate docked to the end of the fusion loop of the envelope protein by forming a conventional hydrogen bond (ARG99), three-carbon hydrogen bonds (ASP98, GLY100, and GLY102), and a Pi donor hydrogen bond (GLY102; Figure 3). The envelope protein-ethyl gallate complex showed a -CDOCKER interaction energy of 22.0534 kcal/mol and binding energy of −5.9868 kcal/mol (Table 1). Moreover, ethyl gallate docked to NS3, defined as an adenosine triphosphate (ATP) interaction site, by forming two conventional hydrogen bonds (THR201 and ASN417), a carbon hydrogen bond (GLY415), Pi-anion (GLU286), and three alkyl bonds (an alkyl bond with VAL227 and VAL228), as well as a pi-alkyl bond with VAL228 (Figure 4). The NS3-ethyl gallate complex showed a -CDOCKER interaction energy of 63.9422 kcal/mol and binding energy of −247.271 kcal/mol (Table 1). In addition, ethyl gallate docked to the priming loop of RdRp by forming a conventional hydrogen bond (ASN716), carbon hydrogen bond (GLU845), Pi-sulfur (CYS730), Pi-Pi stacked (TRP848), and Pi-alkyl (TRP848) bond. Notably, ethyl gallate formed a metal-acceptor bond with Zn^2+^. The RdRp-ethyl gallate complex showed a -CDOCKER interaction energy of 99.9269 kcal/mol and binding energy of −200.43 kcal/mol. These molecular docking results indicated that ethyl gallate docks to the three major proteins by forming several non-bonded interactions; in particular, ethyl gallate forms stronger bonds with NS3 and RdRp than with the envelope protein.

### 2.4. MD Simulation of the Three Protein-Ethyl Gallate Complexes

To analyze the MD simulation of each protein-ethyl gallate complex, the complexes were solvated according to their shape and size. Each solvated protein-ethyl gallate complex showed a stable total energy and root mean square deviation (RMSD) during the MD simulation (Figure 6a,b). By contrast, the absolute solvent-accessible surface area (SASA) of the envelope protein-ethyl gallate complex exhibited a progressive increase relative to the NS3-ethyl gallate and RdRp-ethyl gallate complexes during the early stages of the simulation, and this high level was sustained until approximately the midpoint of the MD simulation (Figure 6c). Furthermore, relative SASA values were calculated using a normalization approach to account for inherent size differences among amino acid residues. Consistent with the absolute SASA results, the relative SASA profiles also demonstrated that the envelope protein-ethyl gallate complex maintained higher solvent accessibility than the NS3-ethyl gallate and RdRp-ethyl gallate complexes up to the midpoint of the simulation (Figure 6d). Furthermore, the stabilities of the individual protein-ligand complexes during the MD simulation were expressed as three types of RMSDs: the complex structure excluding water molecules (Figure 7a), interaction site between the protein and ethyl gallate (Figure 7b), and ligand only (Figure 7c). The RMSD values of the three complex structures, excluding water molecules, exhibited comparable increases until the initial point of the simulation. Subsequently, the RMSD of RdRp-ethyl gallate remained constant at the end of the simulation, whereas the RMSD of the envelope protein-ethyl gallate complex and NS3-ethyl gallate complex exhibited a sustainable increase (Figure 7a). As demonstrated in Figure 7b, the RMSD of the interaction site between the envelope protein and ethyl gallate increased at the initial point and finally exhibited significant fluctuations at the end of the simulation, resulting in the complete removal of ethyl gallate. Conversely, the RMSDs of the interaction sites between NS3, RdRp, and ethyl gallate demonstrated consistent stability, with no variation (Figure 7b). Moreover, ethyl gallate docked to the envelope protein was agitated in the middle of the simulation, unlike the ethyl gallate docked to the other two proteins (Figure 7c). A subsequent comparison of the stabilities of the individual complexes was conducted by calculating the number of non-bonded interactions and hydrogen bonds for each complex (Figure 8). The envelope protein-ethyl gallate complex exhibited repeat formation and extinction of a few non-bonded interactions, such as the LYS251-seconds oxygen of the ethyl gallate bond, during the entire simulation (Figure 8a and Figure 9a). In contrast, two of the remaining complexes maintained non-bonded interactions during the simulation. Notably, the RdRp-ethyl gallate complex demonstrated a specific number of non-bonded interactions (Figure 8a). In the NS3-ethyl gallate complex, ethyl gallate formed non-bonded interactions with several amino acids, including ARG202 and VAL228 (Figure 9b). In the RdRp-ethyl gallate complex, ethyl gallate maintained its interaction with Zn^2+^ and diverse amino acids, including TRP848, ARG771, and CYS730 (Figure 9c). Furthermore, the NS3-ethyl gallate complex and RdRp-ethyl gallate complex exhibited a specific number of hydrogen bonds in comparison to the envelope protein-ethyl gallate complex during the simulation (Figure 8b). These MD simulation results indicate that ethyl gallate strongly binds and interacts with NS3 and RdRp, whereas it falls apart from the complex with the envelope protein.

### 2.5. Molecular Mechanics Poisson-Boltzmann Surface Area (MM-PBSA) Analysis of the Three Protein-Ethyl Gallate Complexes

MM-PBSA calculations were carried out to compare the relative binding energetics of ethyl gallate across the three protein targets. Representative conformations (10 initial and 10 final conformations) were extracted from each MD trajectory, and the MM-PBSA procedure was applied to these conformations to evaluate energetic trends rather than absolute binding free energies. As shown in Table 2 and Figure 10, in the envelope protein, both the initial (−5.6446 ± 11.4089 kcal/mol) and final (−12.9032 ± 9.4303 kcal/mol) sets yielded unfavorable interaction energies, which aligns with the unbinding behavior in molecular docking and MD simulation analysis. In NS3, the initial and final conformations produced consistently favorable energetic values (−38.9574 ± 9.9201 and −38.4431 ± 5.0091 kcal/mol, respectively), supporting the stable binding mode maintained throughout the simulation. In RdRp, the initial conformations showed favorable MM-PBSA energies (−33.8089 ± 12.5282 kcal/mol) comparable to those of NS3. However, the final conformations exhibited substantially less favorable values (−11.9039 ± 20.6951 kcal/mol). This energetic shift corresponds to the structural expansion observed in the RdRp-ethyl gallate complex (Figure 10). Moreover, this structural expansion is not observed in the apo RdRp (See Appendix A for details). These results suggested that ethyl gallate binding may reduce the stability of the protein during the later stages of the simulation. Because MM-PBSA provides an enthalpy-dominated, implicit-solvent approximation, the resulting values should be interpreted in a comparative context. Nonetheless, the relative energetic patterns observed across the three proteins are consistent with the binding behaviors identified in the MD simulations.

## 3. Discussion

Ethyl gallate is a flavonoid and natural antioxidant found in various fruits, vegetables, nuts, plants, and even algae. *Spirogyra* sp., a freshwater green alga, contains phenolic compounds including ethyl gallate, which show protective effects against apoptosis induced by ultraviolet B [14]. Peng et al. [15] announced that various macroalgae contain phenolic compounds, and the amount of ethyl gallate is correlated with total ferric-reducing antioxidant power and 2,2′-azino-bis(3-ethylbenzthiazoline)-6-sulfonic acid activities. Moreover, ethyl gallate possesses significant anti-inflammatory, antioxidant, and antiviral activities, making it a potential candidate for therapeutic applications in various diseases, including inflammation, cancer, and viral diseases [16,17]. The antioxidant properties of ethyl gallate reportedly regulate viral replication and reduce hepatitis C virus [18]. In the present study, we demonstrated the antiviral activity of ethyl gallate in ZIKV-infected Vero E6 cells in vitro. Further, structural analyses using molecular docking and MD simulation provided further insights into the mechanisms underlying its antiviral action.

Viral infection triggers a continuous battle between host defense mechanisms and the replicating virus, in which the IFN response plays a pivotal role as a key component of the innate immune defense against pathogens. In the complex IFN response, the induction of ISGs is essential in shaping antiviral defense mechanisms [19]. One family of ISGs is the IFN-induced transmembrane proteins (IFITMs), including IFITM1, IFITM2, and IFITM3, which limits the viral infection by various viruses [20]. IFITM proteins inhibit the early stages of the viral life cycle by blocking viral entry or intracellular trafficking of viral particles [21]. As shown in Figure 2a,b, in ZIKV-infected cells, the elevated expression of IFITM genes was markedly reduced upon ethyl gallate treatment, suggesting that this effect may be linked to the role of IFITM proteins in restricting viral entry. IFN-induced proteins with tetratricopeptide repeats (IFITs) interact with RNA and cellular proteins to restrict viral translation and modulate innate immune signaling, cell death, and inflammation [22]. Ethyl gallate significantly suppressed the ZIKV-induced upregulation of IFIT genes, indicating that its antiviral efficacy may be associated with the ability of IFIT proteins to restrict viral protein translation, thereby impeding viral replication (Figure 2c–f). *OAS1–OAS3* bonded to double-stranded RNA and catalyzed the synthesis of 2–5As, thereby activating RNase L. Activated RNase L degrades cellular and viral RNAs, ultimately inhibiting protein synthesis and halting viral replication [23]. PKR is activated through autophosphorylation, enhancing its kinase activity and leads to the suppression of protein synthesis in virus-infected cells [12]. Herein, we observed a significant decrease in the expression of *OAS* and *PKR* following ethyl gallate treatment in ZIKV-infected cells (Figure 2g–i), suggesting that this may be associated with their roles in viral RNA degradation and inhibition of viral protein synthesis.

Molecular docking and MD simulation are in silico computational methods used to predict the binding affinity of a ligand to a target protein [24]. Molecular docking explored the docking conformation of a ligand into a protein-binding site, and MD simulation predicted the physical movement of atoms and molecules over time. The serial application of these computational methods identified potential drug candidates by simulating and predicting their binding affinities for target proteins. Stably docked poses exhibited minimal conformational changes during MD simulation [25]. Additionally, MM-PBSA analysis is incorporated to further evaluate the energetic characteristics of ethyl gallate binding to three proteins. Additionally, MM-PBSA analysis was incorporated to further evaluate the energetic characteristics of ethyl gallate binding to three proteins. MM-PBSA does not provide absolute thermodynamic free energies; however, it offers an enthalpy-dominated, implicit-solvent approximation that is widely used to assess the relative ranking of ligand affinities [26,27]. Previous studies have also demonstrated that integrated approaches combining molecular docking, MD simulation, and MM-PBSA can yield biologically meaningful interpretations of ligand-protein interactions [28]. In this study, MM-PBSA calculations were applied to representative conformations sampled from the early and late stages of each trajectory to estimate relative binding energetics and to biologically interpret the results obtained from molecular docking and MD simulation.

The ZIKV life cycle can be divided into attachment and entry into host cells, uncoating, replication and gene expression, assembly in host cells, and release from host cells [29,30]. ZIKV consists of several structural proteins, including envelope proteins, and non-structural proteins such as NS3 helicase and NS5 RdRp, which are crucial in the inherent life cycle stage [29]. Therefore, these major proteins have been investigated as targets for the identification of drug candidates [31].

Envelope proteins regulated viral membrane fusion and entry. The fusion loop of the envelope protein is instrumental in initiating the fusion of the viral envelope with the host cell membrane, leading to viral entry [32]; thus, blocking the fusion loop inhibits viral entry. In the envelope protein-ethyl gallate simulation, ethyl gallate was docked to the fusion loop of the envelope protein. However, the interactions between ethyl gallate and fusion loop were not sustained, and ethyl gallate separated from the envelope protein-ethyl gallate complex, reducing the possibility of ethyl gallate exerting its antiviral activity by blocking the envelope protein.

The ZIKV NS3 helicase plays a pivotal role in viral replication by unwinding genomic RNA with a helicase domain located at the C-terminus [33,34]. This domain contains conserved motifs, including the P-loop, that are essential for ATP binding and catalysis [35]. In the molecular docking analysis of ethyl gallate with NS3, ethyl gallate docked to the ATP-binding site with THR201, VAL227, VAL228, GLU286, GLY415, and ASN417. Ethyl gallate most frequently formed non-bonded interactions with AGR202, a key amino acid that constitutes the P-loop of NS3 helicase and interacts with ATP [34]. MD simulations showed that this binding pose remained stable throughout the trajectory, and the MM-PBSA energetic values were consistently favorable in both the early and late conformations. This stable energetic profile supports the persistence of the binding mode observed during MD. Taken together, these results suggest that ethyl gallate may interfere with ATP engagement or hydrolysis in the NS3 helicase, potentially impairing its essential role in RNA unwinding during ZIKV replication.

RdRp is an enzyme that catalyzes the replication of RNA from an RNA template, and Zn^2+^ contributes to the stability of the RdRp structure by interacting with its amino acids, including CYS730 [36]. In addition, the priming loop (residues 721-903) acted as a stacking platform to position the initiating NTP during de novo initiation [37]. In this study, ethyl gallate was observed to dock near both Zn^2+^ and CYS730. Non-bonded interactions with the priming loop, including Zn^2+^, CYS730, ARG771, GLU845, and TRP848, persisted throughout the MD simulation. These docking poses and MD simulation results suggest that ethyl gallate may interfere with structural or functional role of the Zn^2+^-CYS730 coordination and the priming loop. Interestingly, the RdRp-ethyl gallate complex exhibited a markedly less favorable MM-PBSA energies during the later stage of the trajectory, accompanied by an unfolding-like structural deviation that was not observed in the apo RdRp simulation. The persistence of interactions at residues involved in structural stabilization (including Zn^2+^ and CYS730), together with the late-stage energetic deterioration, is consistent with the possibility that ethyl gallate perturbs local structural elements of the enzyme. This pattern suggests that the presence of ethyl gallate may contribute to ligand-induced destabilization of RdRp. Overall, these findings indicate that ethyl gallate binding may reduce RdRp stability and potentially interfere with its functional conformation during the initiation phase of RNA replication. Although the MM-PBSA analysis provided useful comparative insights into the relative binding energetics, more rigorous free-energy approaches—such as alchemical free energy perturbation (FEP) or thermodynamic integration (TI)—would be valuable in future work to quantitatively validate the binding mechanisms and further refine the energetic interpretation of ethyl gallate interactions with the target proteins. Furthermore, commercial kits or standardized assay systems for measuring the enzymatic inhibition of ZIKV NS3 protease/helicase or RdRp activity have not yet been developed. The development of a reliable enzyme assay platform is essential for future research aimed at validating and developing antiviral lead compounds.

In conclusion, we demonstrated that ethyl gallate exerted antiviral effects by downregulating ISGs, including IFITM, IFIT, OAS, and PKR, in ZIKV-infected cells. While additional experimental analyses are required to fully elucidate the precise antiviral mechanism, our molecular docking, MD simulation, and MM-PBSA data suggest that ethyl gallate may inhibit the enzymatic activities of ZIKV NS3 and RdRp. Such inhibition is suggested to interfere with critical steps of the viral life cycle, thereby reducing viral replication and leading to a downstream decrease in ISG expression. Thus, ethyl gallate is a candidate drug for treating ZIKV infections. Further research is needed to explore other gallate derivatives to assess their potential to exert similar or superior antiviral effects against ZIKV.

## 4. Materials and Methods

### 4.1. Cells, Viruses, and Culture Conditions

Vero E6 cells were obtained from American Type Culture Collection (Washington, DC, USA). Vero E6 cells were grown and maintained in Dulbecco’s Modified Eagle Medium (DMEM) supplemented with 10% heat-inactivated fetal bovine serum and 1% penicillin-streptomycin at 37 °C in a humidified atmosphere containing 5% CO_2_. ZIKV was kindly provided by the Korea Centers for Disease Control and Prevention. The viruses were stored as small aliquots at −80 °C until use.

### 4.2. Sample Treatment and Viral Infection

Cells were plated in six-well culture plates at a density of 1 × 10^6^ cells per well and allowed to adhere overnight. The cells were rinsed with phosphate-buffered saline, and 2 mL DMEM supplemented with 2% fetal bovine serum was added. Cells were treated with ethyl gallate at concentrations of 25, 50, 100 and 200 µM and incubated for 2 h at 37 °C. Next, each well was infected with ZIKV at 0.01 MOI. Following 48 h of incubation, the supernatant was used for the plaque assay to determine viral titers, and the cells were used to quantify the mRNA load using quantitative PCR.

### 4.3. Plaque Inhibition Assay

The viral titer was evaluated using a plaque assay. Viral plaque assays were performed as previously described [38]. Briefly, 10-fold dilutions of the cell culture medium were prepared in DMEM. Each cell medium dilution (0.5 mL) was inoculated onto confluent Vero E6 cell monolayers in six-well culture plates. After the cells were infected with ZIKV, cells were incubated for 2 h at 37 °C, and 5% CO_2_ to allow for virus adsorption. Following 5 d of incubation in DMEM/F12 containing 2% oxoid agar (at a ratio of 7:3), the plaques were fixed with 1 mL of 4% formaldehyde, visualized by staining with 0.1% crystal violet, and counted.

### 4.4. RNA Extraction

Cells were seeded onto six-well plates, incubated for 24 h, then treated with 25–200 µM ethyl gallate for 2 h. Total RNA was isolated from the harvested cells using TRIzol reagent (Invitrogen, Carlsbad, CA, USA) according to the manufacturer’s instructions. The RNA pellet was dried, dissolved using Diethyl pyrocarbonate-water, and stored at −80 °C.

### 4.5. RT-PCR Assay

cDNA synthesis was performed with the High-Capacity RNA-to-cDNA kit (Applied Biosystems, Foster City, CA, USA) following the manufacturer’s instructions. The cDNA amplifications were conducted using the QuantStudio 3 Real-Time PCR System (Applied Biosystems) with the following thermal cycling conditions: 50 °C for 2 min (reverse transcription), 95 °C for 10 min (enzyme activation), and 40 cycles of amplification (95 °C for 15 s and 60 °C for 60 s). GAPDH-RNA expression was used to normalize the cDNA concentration and reactions were performed in QuantStudio™ Design & Analysis Software v1.5.3 (Applied Biosystems). The comparative C_T_ (ΔΔC_T_) method was used to calculate the relative target quantity in gene expression as determined from quantitative reverse transcription PCR experiments. A list of the primers used in this study is provided in Table 3.

### 4.6. Statistical Analysis

All data are expressed as the mean ± standard deviation of three determinations. Statistical comparisons were performed using GraphPad Prism software version 10 (GraphPad Software, San Diego, CA, USA). Differences between groups were evaluated using one-way analysis of variance (ANOVA) followed by Dunnett’s multiple comparison test. A *p* value of <0.05 was considered statistically significant.

### 4.7. Three-Dimensional (3D) Structure of Ethyl Gallate

The structure of ethyl gallate was obtained from PubChem (CID: 13250). The 3D structure of ethyl gallate was prepared and optimized in Discovery Studio 2025 (Biovia, San Diego, CA, USA) through ligand preparation, energy minimization, and conformational analysis workflows.

### 4.8. Acquisition and Processing of ZIKV Protein Structures

For the molecular docking experiments, the 3D structures of the main ZIKV proteins were downloaded from the Protein Data Bank (PDB). The envelope protein (PDB ID: 5JHM), NS3 (PDB ID: 5GJB), and RdRp (PDB ID: 5TFR) exhibited resolutions of 2.00, 1.70, and 3.05 Å, respectively. The 3D structures were prepared using protein preparation protocols and optimized using MD simulation protocols in Discovery Studio 2025. Based on previously reported structural and docking studies, the binding sites of the envelope protein, NS3, and RdRp were defined as follows. For the envelope protein, the binding site was defined around the fusion loop, which forms a conserved hydrophobic motif responsible for pH-dependent membrane fusion with the host cell membrane during virus entry [39]. The docking grid for NS3 was centered on the ATP-interacting region within the helicase domain, corresponding to the site that accommodates ATP during NTP binding and hydrolysis [34]. The binding site of RdRp has been defined to include a priming loop region located above the active site cavity, which plays a crucial role in stabilizing the initiating NTP during de novo RNA synthesis [40,41].

### 4.9. Molecular Docking Analysis Between Proteins and Ethyl Gallate

Molecular docking analysis was performed to assess the structural effects of ethyl gallate on the three main proteins using the CHARMM-based DOCKER (CDOCKER) protocol and Calculate Binding Energies tools in Discovery Studio 2025. The sphere of the active site was extended to a radius of 9.5, 15, and 6.2 Å considering the distance from ethyl gallate and empty space of each protein. The docking poses of ethyl gallate on the three main proteins were expressed as 2D and 3D crystal structures.

### 4.10. Molecular Dynamic Simulations

MD simulations were performed to evaluate the dynamic behavior of each complex of the three main proteins and ethyl gallate using the CHARMM force field in Discovery Studio 2025. Simulation trajectories of protein-ligand complexes were analyzed using simulation functions such as RMSD, SASA, non-bonding, and hydrogen bonding.

### 4.11. MM-PBSA Calculations

MM-PBSA calculations were performed as an implicit-solvent, post-processing approach to estimate the relative binding energetics of the three protein-ethyl gallate complexes. All calculations were carried out in Discovery Studio 2025 using the CHARMM force field. To compare the energetic profiles at different stages of the simulations, the representative conformations were extracted from both the early (initial) and late (final) regions of each MD trajectory. A total of 10 conformations from the initial segment and 10 from the final segment were selected for each complex, and the MM-PBSA procedure was applied to these conformations to evaluate the relative energetic trends rather than absolute binding free energies.

## Figures and Tables

**Figure 1 ijms-26-12062-f001:**
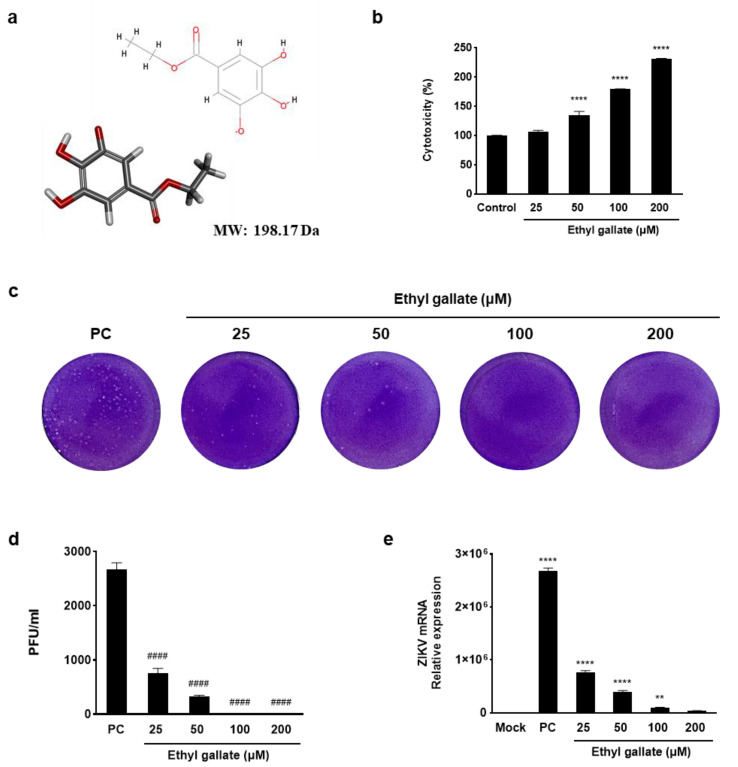
Antiviral activity of ethyl gallate in Zika virus-infected Vero E6 cells. The two−dimensional (2D) and three−dimensional (3D) chemical structures of ethyl gallate (**a**) (oxygen atoms: red; carbon atoms: gray; hydrogen atoms: light gray) and cytotoxicity of ethyl gallate in Vero E6 cells (**b**). Plaque image (**c**) and titer (**d**) on antiviral activity of ethyl gallate using a plaque assay. Effect of ethyl gallate on Zika virus NS1 mRNA expression level (**e**). The data represent mean ± standard deviation from triplication. ** *p* < 0.002, **** *p* < 0.0001 as compared to the Mock group. ^####^
*p* < 0.0001 as compared to the positive control (PC) group.

**Figure 2 ijms-26-12062-f002:**
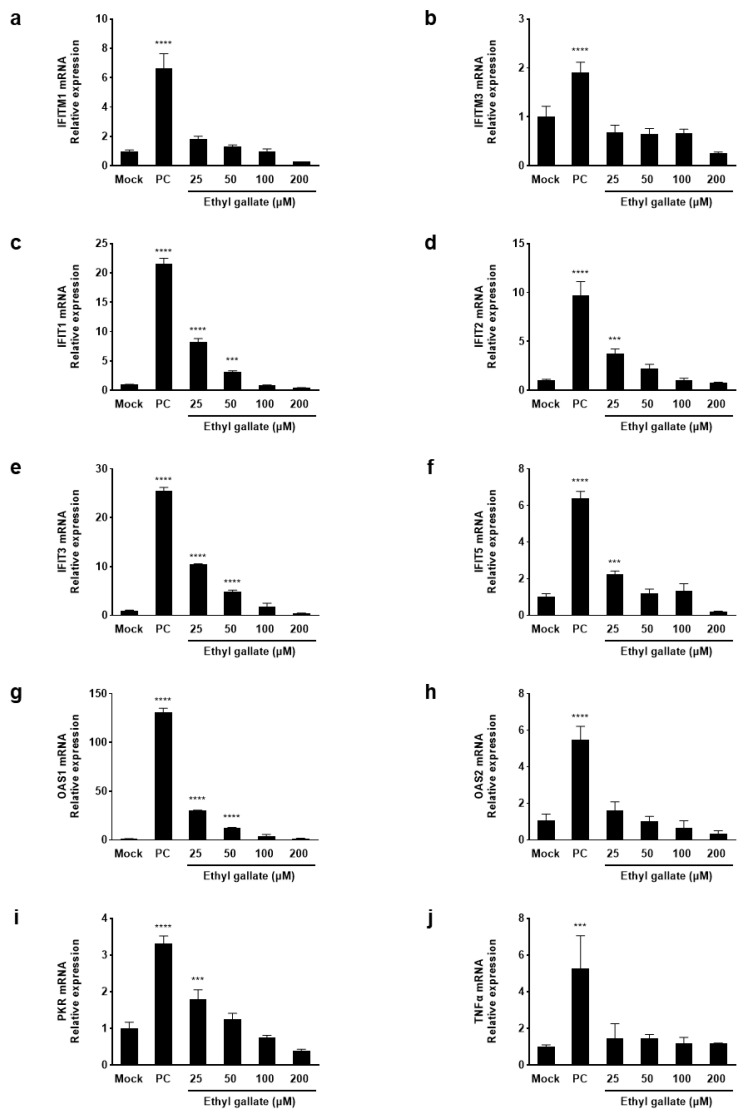
Effects of ethyl gallate on interferon-induced mRNA levels (**a**–**i**), and a pro-inflammatory cytokine, TNF α (**j**) in Zika virus-infected Vero E6 cells. *** *p* < 0.0002, **** *p* < 0.0001 as compared to the Mock group.

**Figure 3 ijms-26-12062-f003:**
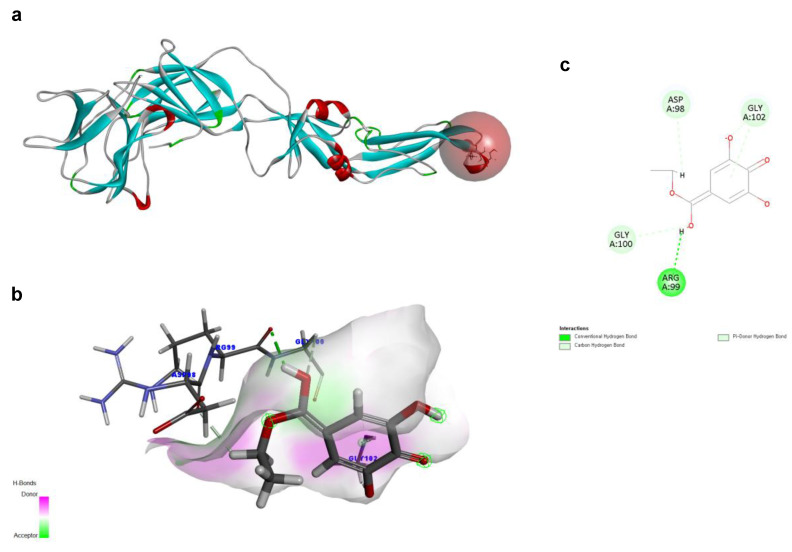
Docking poses of ethyl gallate to the envelope protein of the Zika virus. The three-dimensional (3D) whole structure of the docked complex (**a**). α-Helices are rendered in red, β-sheets in cyan, and the red sphere indicates the active site where the ligand interacted. The 3D structure of the interacting part of the docked complex (**b**). The 2D ligand interaction diagram (**c**).

**Figure 4 ijms-26-12062-f004:**
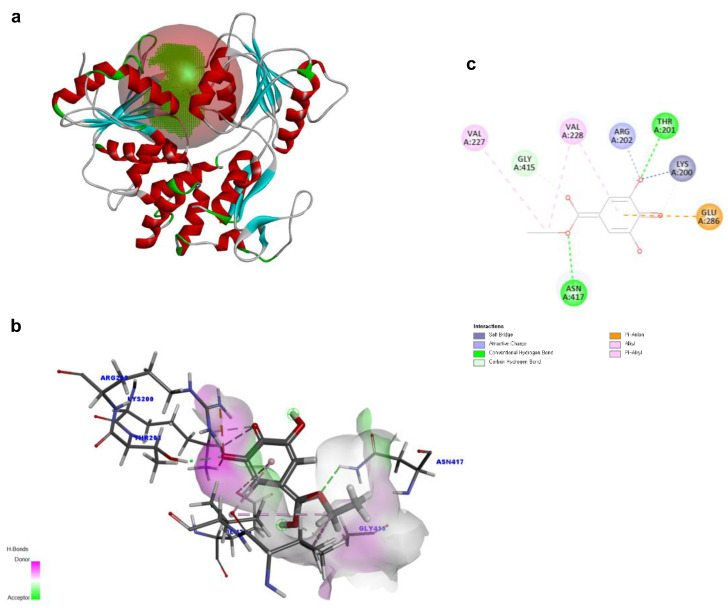
Docking poses of ethyl gallate to the NS3 of the Zika virus. The three-dimensional (3D) whole structure of the docked complex (**a**). α-Helices are rendered in red, β-sheets in cyan, and the red sphere indicates the active site where the ligand interacted. The 3D structure of the interacting part of the docked complex (**b**). The 2D ligand interaction diagram. Among the residues interacting with the ligand, only LYS200 forms two distinct electrostatic interactions—an attractive charge and a salt bridge (**c**).

**Figure 5 ijms-26-12062-f005:**
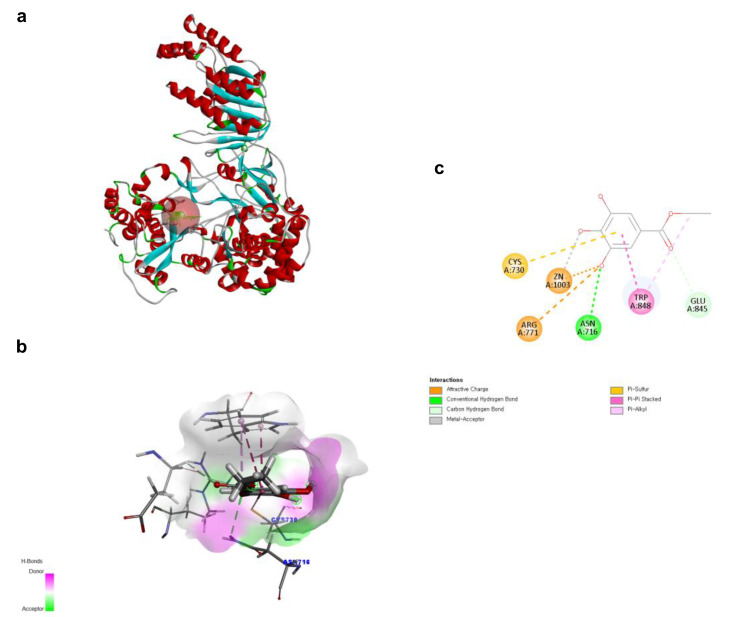
Docking poses of ethyl gallate to the RdRp of the Zika virus. The three-dimensional (3D) whole structure of the docked complex (**a**). α-Helices are rendered in red, β-sheets in cyan, and the red sphere indicates the active site where the ligand interacted. The 3D structure of the interacting part of the docked complex (**b**). The 2D ligand interaction diagram (**c**).

**Figure 6 ijms-26-12062-f006:**
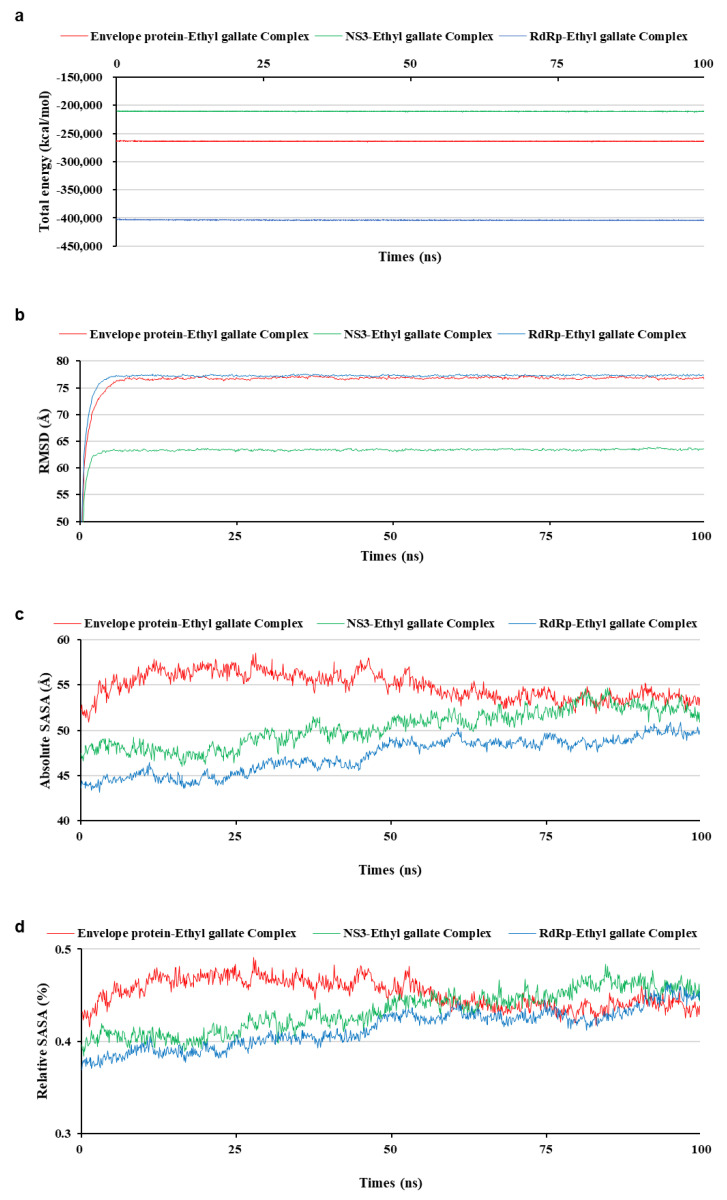
Stability of each solvated whole complex during molecular dynamic simulation. Total energy (**a**), root mean square deviation (**b**), absolute solvent-accessible surface area (SASA) (**c**), and relative SASA (**d**).

**Figure 7 ijms-26-12062-f007:**
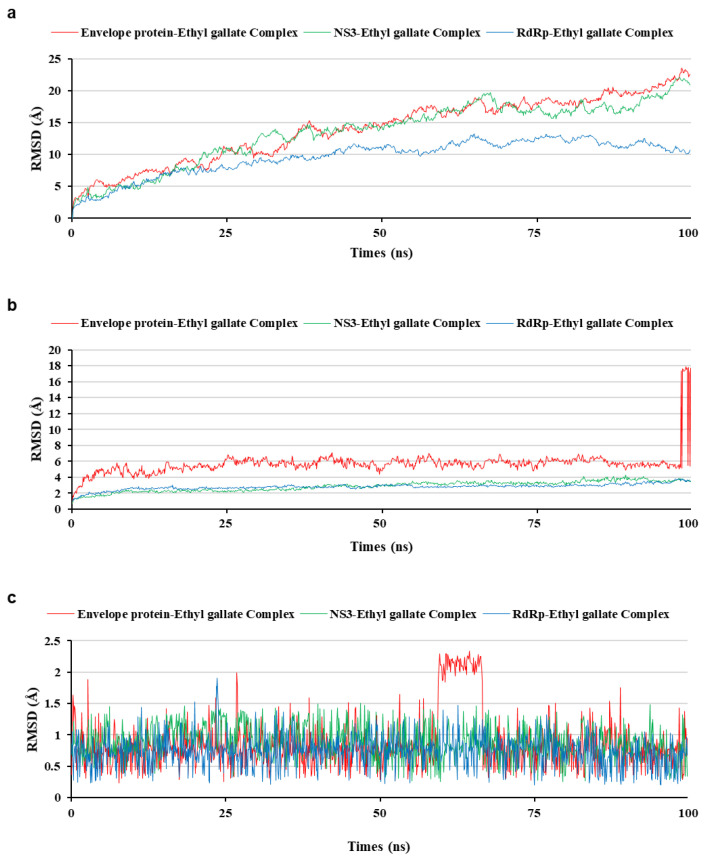
Comparison of the root mean square deviation (RMSD) of three protein-ethyl gallate complexes during molecular dynamic simulation. RMSD on the whole complex structure without water molecules (**a**), on the interaction site between the protein and ethyl gallate (**b**), and ethyl gallate alone (**c**).

**Figure 8 ijms-26-12062-f008:**
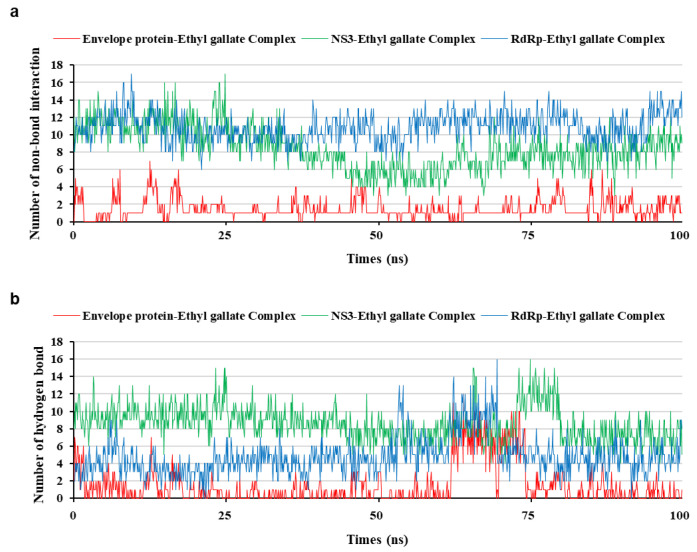
Interaction bond comparison of three protein-ethyl gallate complexes during molecular dynamic simulation; non-bond interaction (**a**) and hydrogen bond (**b**).

**Figure 9 ijms-26-12062-f009:**
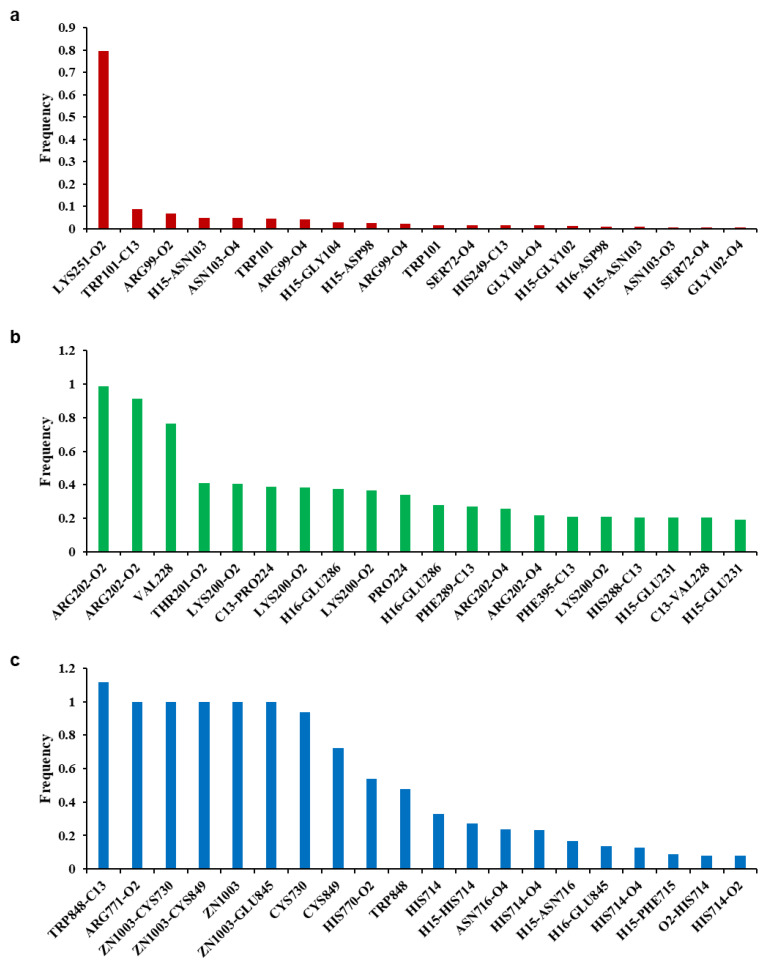
Frequency of non-bonded interactions during molecular dynamic simulation. Envelope protein-ethyl gallate (**a**), NS3-ethyl gallate (**b**), and RdRp-ethyl gallate (**c**).

**Figure 10 ijms-26-12062-f010:**
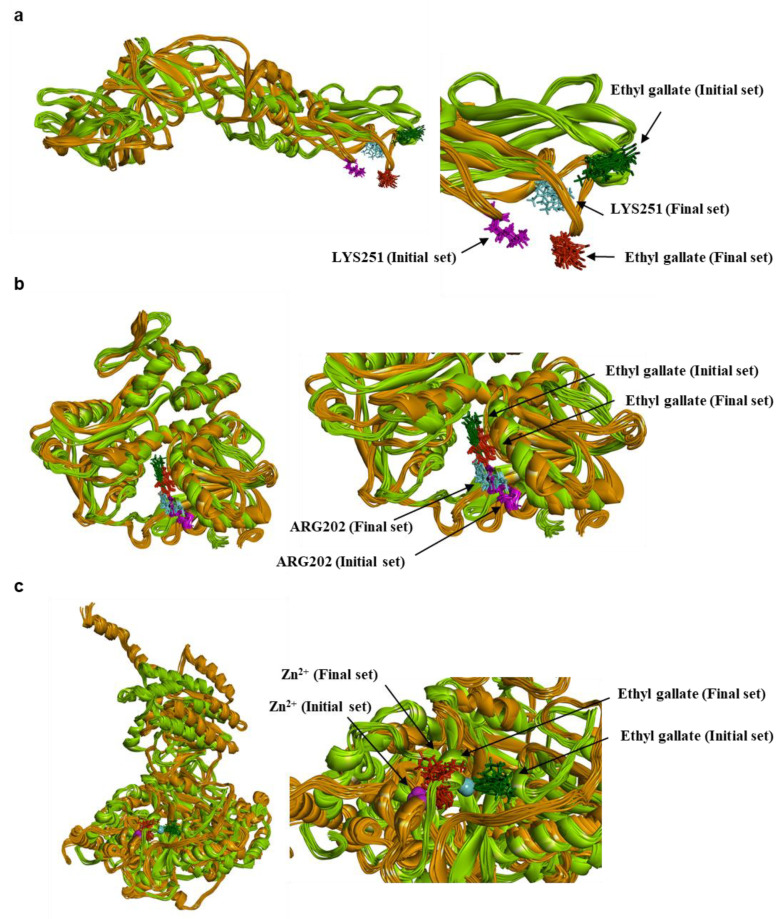
Conformational change during MD simulation. Envelope protein-ethyl gallate (**a**), NS3-ethyl gallate (**b**), and RdRp-ethyl gallate (**c**). Green color family (protein: light/forest green; ethyl gallate: forest green; residues: cyan); Initial 10 conformations, Orange color family (protein: orange; ethyl gallate: red; residues: pink); Final 10 conformations.

**Table 1 ijms-26-12062-t001:** Calculated binding energies of ethyl gallate to the three main Zika virus proteins.

	-CDOCKER Interaction Energy (kcal/mol)	Binding Energy (kcal/mol)
Envelope protein	22.0534	−5.9868
NS3	63.9422	−247.271
RdRp	99.9269	−200.43

**Table 2 ijms-26-12062-t002:** Estimated relative binding energetics from Molecular mechanics Poisson-Boltzmann surface area (MM-PBSA) calculation for the three protein-ligand complexes.

Complex	Stage	Number of Frame	Relative MM-PBSA Energy (kcal/mol)	Interpretation
Envelope protein-ethyl gallate	Initial	10	−5.6446 ± 11.4089	Weak binding/unstable
Envelope protein-ethyl gallate	Final	10	−12.9032 ± 9.4303	Still weak; consistent with unbinding
NS3-ethyl gallate	Initial	10	−38.9574 ± 9.9201	Strong binding
NS3-ethyl gallate	Final	10	−38.4431 ± 5.0091	Stable strong binding
RdRp-ethyl gallate	Initial	10	−33.8089 ± 12.5282	Favorable; stable pose
RdRp-ethyl gallate	Final	10	−11.9039 ± 20.6951	Greatly weakened; consistent with unfolding

The relative MM-PBSA energies (kcal/mol), presented as mean ± SD, were estimated from representative conformations sampled from the initial and equilibrated final stages of the MD trajectories. More negative values indicate more favorable ligand-protein binding.

**Table 3 ijms-26-12062-t003:** Primer sequences used for RT-PCR.

Primer Name		Primer Sequence
GAPDH	Forward	5′-GCA AAT TCC ATG GCA CCG T-3′
	Reverse	5′-TCG CCC CAC TTG ATT TTG G-3′
Zika virus NS1	Forward	5′-CRA CTA CTG CAA GYG GAA GG-3′
	Reverse	5′-GCC TTA TCT CCA TTC CAT ACC-3′
IFITM1-B	Forward	5′-GGG CAT CCT CAT GAC CAT TGG A-3′
	Reverse	5′-GGC TAC TAG TAA CCC CGT TTT TCC TG-3′
IFITM3-B	Forward	5′-ACC ATG AAT CAC ACT GTC CAA ACC TT-3′
	Reverse	5′-CCA GCA CAG CCA CCT CG-3′
IFIT1	Forward	5′-ACA CCT GAA AGG CCA GAA TG-3′
	Reverse	5′-GCTTCTTGCAAATGTTCTCC-3′
IFIT2	Forward	5′-ATC CCC CAT CGC TTA TCT CT-3′
	Reverse	5′-CCA CCT CAA TTA ATC AGG CAC T-3′
IFIT3	Forward	5′-AGG AAG GGT GGA CAC AAC TG-3′
	Reverse	5′-TGG CCT GTT TCA AAA CAT CA-3′
IFIT5	Forward	5′-CGT CCT TCG TTA TGC AGC CAA G-3′
	Reverse	5′-CCC TGT AGC AAA GTC CCA TCT G-3′
OAS1	Forward	5′-GAT CTC AGA AAT ACC CCA GCC A-3′
	Reverse	5′-AGC TAC CTC GGA AGC ACC TT-3′
OAS2	Forward	5′-AGA AGC TGG GTT GGT TTA TC-3′
	Reverse	5′-GAC GTC ACA GAT GGT GTT C-3′
PKR	Forward	5′-ACG CTT TGG GGC TAA TTC TT-3′
	Reverse	5′-TTC TCT GGG CTT TTC TTC CA-3′
TNFα	Forward	5′-CCA ACT GTC ACT CAT TGC TGA-3′
	Reverse	5′-TTC CAA GAA GGA GAC CAT GTT T-3′

## Data Availability

The original contributions presented in this study are included in the article. Further inquiries can be directed to the corresponding author.

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
