# Peer review of "Antiviral Activity of Ethyl Gallate Against Zika Virus: In Vitro and In Silico Studies"

_ijms, 2025, doi:10.3390/ijms262412062_

Round 1

Reviewer 1 Report

Comments and Suggestions for Authors

In this manuscript, the authors present integrated computational and functional results of ethyl gallate against ZIKV. While the premise of the work is potentially interesting, much more details need to be comprehensively provided.

  1. Figure 1 shows the structure of ethyl gallate. Does this consider sidechain rotamers or is it unique? Why?
  2. Does flexible vs rigid docking affect the results? How were the docking poses obtained evaluated for reproducibility? Why is the active site unique? How does stochasticity in the active site definition affect the docking results?
  3. The authors need to comprehensively explain and quantitatively evaluate the effects/redundancy of sidechain rotamers of the docked molecule. Without this, the results could easily be biased.
  4. How does the 2D ligand interaction diagram depend on sidechain flexibility?
  5. The results from MD simulations is very confusing. How many replicates were performed? Why does the total energy is a "straight line"? This seems very strange. 
  6. Can the authors show plots for internal energy and kinetic energy obtained from each MD trajectory?
  7. How were RMSD values calculated? Why no sidechain atoms were taken into account to calculate simulated RMSD values?
  8. The solvent accessibility involves relative and absolute SASA values. The authors need to provide plots for both of these SASA measures.
  9. It seems surprising that no energetics/free energy calculations of the docked poses were performed. Only contact/bond frequency formation only provides superficial metric of the simulated trajectories. Energetics obtained within MD trajectories needs to be quantitatively described.

Reviewer 2 Report

Comments and Suggestions for Authors

Viral diseases are a public health burden, accounting for more than half of infectious diseases worldwide. A lot of severe diseases are caused by RNA viruses, such as influenza, AIDS, hepatitis A, C, D, E, and G, measles, poliomyelitis, Ebola hemorrhagic fever, etc. Zika virus is in the same group and currently has pandemic status. The search for new antiviral agents against Zika virus remains relevant. From this point, the work "Antiviral activity of ethyl gallate against Zika virus…" by Yeon-Ji Lee et al. seems very useful and is of current interest.

The authors investigated the inhibitory effects of ethyl gallate against Zika virus using antiviral activity evaluation, molecular docking, and molecular dynamic simulations. They found that ethyl gallate caused dose-dependent suppression of viral infection without inducing cytotoxicity, inhibited the increase in the expression of interferon-stimulated genes in infected cells. Molecular dynamics simulation showed that the ethyl gallate - RdRp complexes were more stable than those of the ethyl gallate enveloped protein complex.

Relevant methods were used to solve the problem. Results of the study are convincing. The article is well and clearly written. A sufficient number of relevant articles have been cited. The authors provided the required level of statistical analysis of the data found. Results are well discussed and compared with known data.

However, to improve the quality of the manuscript, it would be advisable to address the following issue. The introduction section should be expanded by including the data on antiviral drugs used to treat Zika disease.

After minor revision, the manuscript can be accepted for publication.

Reviewer 3 Report

Comments and Suggestions for Authors

The authors of the manuscript found that general ethyl gallate behave anti Zika bioactivity through cell experiment and molecular docking. Of course, the discovery of the manuscript help Zika associated drug discovery. Therefore, the creativity of the manuscript is super enough for the acceptance by the International Journal of Molecular Sciences. Here are some concerns for the authors to improve the quality of the manuscript:

1) In the Keywords section, the “interferon-stimulated genes” should be deleted, and “molecular dynamic simulation” should be added after molecular docking. The right words sequence should be: Zika infection; ethyl gallate; antiviral; molecular docking; molecular dynamic simulation.

2) The first two sentences of the Introduction section should be supported by references.

3) In section 2.1, for Figure 1a, the 2D chemical structure should be added. In line 58, the authors should check the unit [MOL].

4) In the whole manuscript, the authors used qPCR or other methods to present “reverse transcription PCR (RT-PCR)” whether in the Results or the Methods sections. The authors should unify writing style as “RT-PCR” for better reading.

5) In section 2.3 and the Abstract section, high binding energies (-247 and -200) were calculated. The authors should re-calculate these two binding energies.

6) For Figure 3, 4, and 5, the Figure b and c indicate the 2D and 3D, respectively. However, the legends were written in different sequences. The authors should recheck these three figures and legends.

Reviewer 4 Report

Comments and Suggestions for Authors

The manuscript presents a well-structured investigation into the antiviral potential of ethyl gallate against the Zika virus (ZIKV). The study integrates in vitro experiments with computational simulations to elucidate mechanisms of action. The findings are novel and contribute significantly to the field of antiviral drug discovery. However, several issues require clarification and enhancement.

Please clarify the rationale for selecting the ethyl gallate concentration range (25–200 µM) and justify whether these doses are physiologically relevant.

It is essential to include details regarding the positive control (e.g., known ZIKV inhibitors) to benchmark the efficacy of ethyl gallate.

While the downregulation of interferon-stimulated genes (ISGs) by ethyl gallate is intriguing, it lacks direct mechanistic connections. Specifically, how does ethyl gallate suppress ISG expression without compromising host immunity?

The molecular docking and molecular dynamics (MD) simulations suggest that NS3 and RdRp are primary targets. Please validate these predictions through enzymatic assays (e.g., ATPase/helicase activity for NS3 and RNA synthesis for RdRp).

Ensure that all quantitative data (e.g., plaque counts and mRNA levels) are reported alongside their statistical significance (e.g., p-values) across all figures.

The MD simulation data indicate instability in the envelope protein complex while showing stability with NS3/RdRp. Please elaborate on why the interaction with the envelope protein fails and discuss whether this affects viral entry inhibition.

Round 2

Reviewer 1 Report

Comments and Suggestions for Authors

While I appreciate the authors' efforts to constructively address the key questions raised by this reviewer, the key analysis regarding true energetics (which is not internal energy and/or kinetic energy) have not been performed. By true energetics, I meant physiologically connecting and assessing the reliability of the docked poses by evaluating binding free energies.

Reviewer 3 Report

Comments and Suggestions for Authors

The authors have revised the manuscript according to the concerns. And the present version of the manuscript is recommended to be accepted by the journal IJMS now. Congratulations to the authors.

Author Response

Comments 1: The authors have revised the manuscript according to the concerns. And the present version of the manuscript is recommended to be accepted by the journal IJMS now. Congratulations to the authors.

Response 1: We sincerely thank the reviewer for the positive evaluation. We appreciate your constructive comments throughout the review process, which greatly improved the quality and clarity of our manuscript.

Reviewer 4 Report

Comments and Suggestions for Authors

Thank you for the revision, the quality of the manuscript has improved significantly.

Author Response

Comments 1: The authors have revised the manuscript according to the concerns. And the present version of the manuscript is recommended to be accepted by the journal IJMS now. Congratulations to the authors.

Response 1: We sincerely appreciate the reviewer’s positive feedback. Thank you for your valuable comments and constructive guidance throughout the review process.

Round 3

Reviewer 1 Report

Comments and Suggestions for Authors

In the second revised version, the authors have substantially attempted to address the pertinent questions raised in the previous round of review.